# Dyslexia, the Amsterdam Way

**DOI:** 10.3390/bs14010072

**Published:** 2024-01-19

**Authors:** Maurits W. van der Molen, Patrick Snellings, Sebastián Aravena, Gorka Fraga González, Maaike H. T. Zeguers, Cara Verwimp, Jurgen Tijms

**Affiliations:** 1Developmental Psychology, Department of Psychology, University of Amsterdam, 1018 WS Amsterdam, The Netherlands; 2Rudolf Berlin Center for Learning Disabilities, University of Amsterdam, 1018 WS Amsterdam, The Netherlands; 3RID, 6811 AJ Arnhem, The Netherlands; 4Center for Reproducible Science, University of Zürich, 8001 Zürich, Switzerland; 5Samenwerkingsverband VO Amsterdam-Diemen, Bijlmermeerdreef 1289, 1103 TV Amsterdam, The Netherlands

**Keywords:** dyslexia, assessment, intervention, EEG, event-related brain potential, graph analysis, machine learning, network analysis, diffusion modelling

## Abstract

The current aim is to illustrate our research on dyslexia conducted at the Developmental Psychology section of the Department of Psychology, University of Amsterdam, in collaboration with the nationwide IWAL institute for learning disabilities (now RID). The collaborative efforts are institutionalized in the Rudolf Berlin Center. The first series of studies aimed at furthering the understanding of dyslexia using a gamified tool based on an artificial script. Behavioral measures were augmented with diffusion modeling in one study, and indices derived from the electroencephalogram were used in others. Next, we illustrated a series of studies aiming to assess individuals who struggle with reading and spelling using similar research strategies. In one study, we used methodology derived from the machine learning literature. The third series of studies involved intervention targeting the phonics of language. These studies included a network analysis that is now rapidly gaining prominence in the psychopathology literature. Collectively, the studies demonstrate the importance of letter-speech sound mapping and word decoding in the acquisition of reading. It was demonstrated that focusing on these abilities may inform the prediction, classification, and intervention of reading difficulties and their neural underpinnings. A final section examined dyslexia, conceived as a neurobiological disorder. This analysis converged on the conclusion that recent developments in the psychopathology literature inspired by the focus on research domain criteria and network analysis might further the field by staying away from longstanding debates in the dyslexia literature (single vs. a multiple deficit, category vs. dimension, disorder vs. lack of skill).

## 1. Introduction

The aim of this paper is to present a concise review of our dyslexia research within the context of the Rudolf Berlin Center at the University of Amsterdam. Rudolf Berlin was a German ophthalmologist who coined the term ‘*dyslexia*’ in his monograph ‘*Eine besondere Art der Wortblindheit (Dyslexie)*’ [1]. Howell [2] argues, however, that Berlin first introduced the term ‘dyslexia’ in presentations he made to two different professional audiences in 1883. References to excerpts of these presentations can be found in Howell’s short note.

Berlin’s use of the term ‘*dyslexia*’ refers to a difficulty with reading that he thought to originate most likely from a brain dysfunction in the left hemisphere [3]. Berlin’s analysis of a modest number of case-studies led him to characterize dyslexia as a pronounced difficulty with reading that is more prevalent in males than females. He suggested that dyslexia cannot be attributed to known disturbances of eyesight or other physical diseases. Further, it is not to be associated with “hebetudo”, a general lethargy or dullness. The afflicted individual does exhibit a strong feeling of distress in the absence of any physical discomfort.

Berlin noted that dyslexia does not seem to have any precursor and observed that its course is varied—sometimes reading improves, but, most frequently, the prognosis is unfavorable. Typically, dyslexia will grow into a serious brain disease. When reading is seen to improve, he assumed that other brain areas took over the functions of the disease-damaged ‘*reading center*’.

Inspired by Berlin’s seminal examination of dyslexia, our research seeks to obtain a deeper understanding of reading difficulty, advance new tools for diagnosis and remediation, and examine its neural correlates. The current review consists of four sections. First, we will briefly present the context of our research program on dyslexia (Section 2). We will then summarize our conceptual starting point, which is followed by illustrations of our research aimed to provide a deeper understanding of dyslexia (Section 3) and illustrations of our research strategies in the assessment (Section 4) and intervention of dyslexia (Section 5). A brief summary of the findings that emerged from our research, together with a consideration of dyslexia as a neurobiological concept, will close the current review (Section 6).

## 2. Context

Before reviewing the findings that emerged from our research, it is in order to provide some context to the research of dyslexia and reading difficulties in the Netherlands. Bibliometric analyses can be used to obtain an estimate of the international status of dyslexia research in the Netherlands. Huang and co-workers amassed close to 20,000 educational-related publications and observed that The Netherlands ranks among the top five countries in educational research [4]. More specifically, Wu and colleagues noted that The Netherlands is within the top 10 most productive countries in terms of dyslexia publications [5]. These authors also analyzed the collaborative ties of the 50 most productive organizations. The outcome of this analysis is graphically presented in Figure 1. In this figure, the size of the nodes represents the number of publications, and the lines between nodes indicate collaborative frequency. Three Dutch universities can be seen at the left (i.e., the University of Amsterdam, the University of Groningen, and Radboud University). A Web of Science search with “*Dyslexia*” and “*Netherlands*” as key-entries returned a total of 545 publications with an H-index of 57. The top 10 most productive authors are listed in Table 1. The top-cited author is Leo Blomert (University of Maastricht). His seminal work on the integration of letters and speech sounds and the neural processes activated by this integration has been an important source of inspiration for our studies [6].

At the applied level, the dyslexia protocol of the Netherlands Quality Institute Dyslexia (NKD) offers a fairly clear picture of the national dyslexia context [7]. The protocol provides guidelines for the diagnosis and remediation of dyslexia. Dyslexia is conceived as a multicausal developmental disorder resulting from an interplay of biological and environmental factors and showing different degrees of manifest difficulty with reading and spelling. At the cognitive level, difficulties in the processing of phonological information are thought to be most prominent. Atypical development resulting in phonological deficits and difficulties in reading and spelling are considered to be manifest already before the initiation of institutional reading and spelling instruction. Considering the negative impact of dyslexia on individual functioning and wellbeing and the societal burden it carries, effective remediation is indicated. The protocol recommends a “*stepped-care*” remediation policy. That is, remediation should be tailored to the severity of the reading and spelling difficulties and the individual needs of the client. Remediation should be provided to clients with persistent reading and/or spelling difficulties (about 1.5 sd below average) who do not suffer from general learning problems, broad neurobiological disturbances, or serious sensory limitations. Diagnosis and remediation should acknowledge frequent comorbidity (e.g., ADHD) of dyslexia with other disorders. Remediation should be evidence-based and inspired by the understanding that phonological deficits are most prominent in dyslexia. The protocol indicates that, on average, clients will need between 45 and 65 h of focused remediation. Remediation can be concluded when reading and spelling are within 1 sd of the average performance. Finally, the protocol provides detailed specifications of diagnostic and remediation procedures, the qualifications of care providers, the embedment of dyslexia care in educational and clinical environments, and insurance regulations.

## 3. Understanding

Our understanding of dyslexia was initially guided by the definition of dyslexia of the International Dyslexia Association (IDA):


*“Dyslexia is a specific learning disability that is neurobiological in origin. It is characterized by difficulties with accurate and/or fluent word decoding and by poor spelling and decoding abilities. These difficulties typically result from a deficit in the phonological component of language that is often unexpected in relation to other cognitive abilities and the provision of effective classroom instruction. Secondary consequences may include problems in reading comprehension and reduced reading experience that can impede growth of vocabulary and background knowledge” [8].*


The above diagnosis emphasizes the importance of phonology and points to the neurobiological origin of dyslexia. Both phonology and neurobiology are key in our studies of dyslexia. Within this framework, reading problems in children diagnosed with dyslexia manifest themselves most prominently in extreme difficulties in acquiring basic reading skills such as the recognition of written words and letter-speech sound integration. But a preliminary cautionary note is in order here. A concise review of the pertinent literature indicates that unraveling the causes of reading problems from their consequences remains a challenge [9]. Nevertheless, this review continues to point to the importance of phonological skills when considering dyslexia and strongly suggests that deficits in phonological processing antecede reading acquisition. Basically, this review echoes the conclusion from an earlier and more substantive review, concluding that phonological deficiencies are the most probable cause of dyslexia while excluding visual, semantic, or syntactic deficits together with general learning problemfs or low-level sensory issues [10].

We focused on two basic phonological skills—letter speech-sound mapping and word decoding. In a relatively transparent orthography, such as Dutch, children acquire letter-speech sound associations within approximately one year of formal reading instruction, but it may take several more years of instruction before these associations are fully automated [11,12]. Numerous studies indicated that letter-speech sound mapping is compromised in dyslexia [13]. It is also acknowledged that impaired word decoding, that is, difficulty identifying real words, is a prominent characteristic of dyslexia, together with the decoding and pronunciation of pseudowords [14]. Difficulties in letter–speech sound mapping and word decoding contribute to what is the most troublesome problem in dyslexia—the inability to read fluently. Reading fluently, accurately, and with good understanding is the hallmark of a skilled reader. This ability is compromised in individuals with dyslexia, having negative, long-term consequences on academic career, employment status, and more general quality of life [15].

The studies reviewed in this section adopted four different research strategies: the reading of an artificial script, diffusion modeling of reaction times, the analysis of event-related brain potentials, and graph analysis of the electroencephalogram (EEG).

### 3.1. Artificial Script Reading

The first study reviewed here examined dyslexia using an artificial script [16]. The use of artificial script has two important advantages over the use of typical script. First, it offers precise control of potential script-exposure differences between individuals with dyslexia and typical readers. Secondly, the use of an artificial script allows for examining the learning rate of letter–speech sound mappings at different ages without the need to use preliterate children with a familial risk for dyslexia [17]. The artificial script consisted of Hebrew graphemes that were matched to Dutch phonemes, thereby providing nine basic letter–speech sound pairs. From the corpus of words that could be created by combining the nine chosen phonemes, we selected 116 high-frequency Dutch monosyllabic words, of which 86 words were used for training purposes. The remaining 30 words were used for the word reading assessment. Thus, by using other words for training purposes rather than for assessment, the transfer of learning could be assessed, ruling out the possibility that words from the training were recognized without decoding. An additional set of 52 pseudowords was constructed that were also used for training purposes. Participants were 62 children diagnosed with dyslexia and 64 children with average or above-average reading skills. The age range was between 7.5 and 12.4 years. A computerized game required children in both groups to match speech sounds to their corresponding orthographic representations. Correct associations contributed to success in the game, whereas incorrect associations reduced a positive outcome. The children played the game for two blocks of 30 min. The results revealed that letter knowledge did not discriminate between groups. In contrast, the children with dyslexia committed more errors during the letter–speech sound mapping task compared to the typical reading children. Moreover, children with dyslexia were less fluent than typical readers when reading the artificial script. Finally, the results showed a significant correlation between the speed of reading an artificial and natural script, suggesting the validity of the artificial script procedure. Collectively, the results obtained by Aravena and co-workers convincingly demonstrate that letter-speed sound mapping difficulties interfere with fluent reading.

Recently, we replicated and extended the key findings derived from the use of the artificial-script task [18]. Participants in this study were 107 elementary school children (age range between 7.7 and 11.3 years). They performed a 30 min version of the artificial-script task. Half of the children received explicit goal-directed instruction informing them about the nature of the task. The other half of the children received implicit instructions indicating that they would play a computer game and that the goal of this game would become evident when playing it. In order to extend our previous findings, all children performed, subsequent to the artificial script task, on a congruency task presenting visual symbol-auditory letter pairs of the artificial script task. Half of the pairs were congruent (i.e., these pairs received positive feedback during the artificial script task), while the other pairs were incongruent (i.e., these pairs received negative feedback). It was anticipated that a congruency effect (i.e., faster and, probably, more accurate responses on congruent relative to incongruent trials) would be manifested due to symbol–letter sound learning when performing the artificial script task. All children took a symbol knowledge task (i.e., a task presenting the symbols learned during the artificial script and asking for the corresponding letter sound). Finally, a word reading task in the artificial script was presented twice; one time immediately following the symbol learning task and a second time after a good night’s sleep. This feature of the study design allowed for examining the consolidation of the symbol–letter sound learning.

The findings showed that the learning rate (in terms of accuracy) was faster for children who had received explicit goal-directed instruction relative to the children who were not informed about the nature of their task. Further, children who received explicit instructions showed better symbol knowledge than children receiving implicit instructions. These findings replicate earlier observations reported by Aravena [16] in children with dyslexia and typical readers and the more recent findings reported by Rastle and co-workers in adult typical readers [19]. In addition, consistent with our previous findings, individual reading rates in reading the artificial script correlated positively with the reading rates of Dutch and pseudowords, thereby supporting the external validity of artificial script learning. The results obtained for the consolidation manipulation were seen to depend on the acquired knowledge of symbol–letter sounds pairings. Children who exhibited greater knowledge were more fluent in reading the artificial script during the second day of testing. The finding that children who received explicit instructions were faster and more efficient in learning the artificial script, together with a better learning outcome than children who were not instructed about the goal of the task, is important as it adds to the growing evidence that top-down control contributes positively to the initial stages of reading acquisition [20,21]. The consolidation effect observed in this study fits in nicely with the findings reported by Wang and co-workers, who observed that even a short nap following the learning or letter–speech sound mappings beneficially affected the performance on a subsequent letter–knowledge task [22]. In a broader context, our consolidation finding is consistent with the well-established positive effect of sleep on memory storage and retrieval [23].

### 3.2. Diffusion Modelling of Reaction Times

Children diagnosed with dyslexia have great difficulties in accurate and, in particular, fluent word decoding [24,25]. Lexicality, the differences between words and pseudowords, is considered one of the hallmarks of the difficulty in word decoding [26,27]. We examined word decoding in dyslexia by using diffusion modeling and electrophysiology. The advantage of diffusion modeling is that it provides a unified account of the speed and accuracy of performance, two performance measures that are typically examined separately. The diffusion model [28] considers the processing captured by two-choice reaction time to consist of a decision process preceded by perceptual and followed by motor processes. The focus of the diffusion model is on the single-staged decision process that is assumed to accumulate noisy information from a starting point towards one of two response criteria or boundaries (word vs. pseudoword in a lexical decision task). A response is emitted when one of the response boundaries is reached. Speed–accuracy tradeoffs occur when the distances between the starting point and boundaries change. Responses are slow when the accumulation of information is slow and/or the distance between the starting point and the corresponding boundary is large and fast when the accumulation rate is high and/or the distance is small. The parameters of the model refer to the starting point of information accumulation (*z*), the accumulation of phonological information as reflected in drift rate (*v*), the response boundaries (labeled *a* and *o*), and the collective processes outside the decision mechanism (*Ter*). A schematic of the diffusion model is presented in Figure 2.

We performed two experiments using diffusion modeling of word decoding [29]. In the first experiment, we used a visual lexical decision task. The sample consisted of 57 children with reading difficulties and 57 nonimpaired readers (age range 9.6 to 11.6 years). They completed a lexical decision task (80 words and 80 pseudowords) derived from Martens and De Jong [30]. A letter string (3 to 5 letters) was presented briefly (120 ms) following a fixation cross with a duration of 250 ms. The termination of the letter string coincided with the presentation of a mask (#####). Participants responded by pressing, as quickly and accurately as possible, one of two keys. Each participant was presented with 480 letter strings—240 words and 240 pseudowords in counterbalanced order. The findings indicated that relative to typical readers, poor readers were less accurate and slower to respond to both words and pseudowords. Diffusion modeling revealed almost perfect fits between actual reaction times and estimated reaction times for both words and pseudo words and both groups of participants. The fits were slightly less perfect for accuracy due to the lower number of observed errors. Anticipated drift rates were lower for impaired readers compared to typical readers, and this finding was similar for words and pseudowords. In addition, boundary separation was larger for impaired readers relative to typical readers. Finally, groups did not differ in *Ter*; that is, the efficiency of processing outside the decision mechanism was similar between groups.

The second experiment comprised three groups of participants—47 children with dyslexia (age range from 9.6 to 11.6 years), 50 age-matched typical readers (age range from 9,7 to 11.4 years), and 36 reading-matched children (age range from 7.1 to 8.8 years). All children performed on a visual lexical decision task, similar to the one used in the first study, and an auditory version of this task in which the stimuli that were used consisted of (pseudo)words read by a female Dutch speech therapist. The results of the visual lexical decision task showed that similar to the data from the first experiment, children with dyslexia were slower and less accurate in responding to both words and pseudowords compared to typical readers. They were faster than reading-matched children, but accuracy did not differ between these two groups. As in the first experiment, diffusion-model estimated reaction times corresponded closely to actual reaction times for both words and pseudowords and in all three groups. A similar pattern was observed for accuracy. Further, the results for the visual lexical decision task replicated the previous findings in showing a lower drift rate for children with dyslexia compared to age-matched typical readers and a wider boundary separation (the latter finding close to significance). Compared to the reading-matched children, the dyslexic children showed less separation between response boundaries and a higher drift rate. Group comparisons did not reveal a difference in *Ter*. Finally, the results from the auditory lexical decision task showed that children with dyslexia had lower drift rates for words and pseudo words compared to age-matched typical readers, but boundary separation did not differ between groups in contrast to *Ter*. In contrast to the findings derived from the visual lexical decision task, processing outside the decision mechanism was, this time, more efficient in dyslexics than age-matched typical readers. The parameters of the diffusion model did not show any difference between children with dyslexia when compared to reading-matched children.

Overall, the findings that emerged from this study are in agreement with research showing that visual word decoding by individuals with dyslexia is both slow and inaccurate [31,32]. Our diffusion modeling added greater precision. The results that emerged from this modeling demonstrated that the lexicality difficulties experienced by dyslexics are accounted for primarily by a delay in the linguistic decision processes that extract information about word characteristics. Additionally, the modeling revealed that the lexicality difficulties are exacerbated by the adoption of wider response boundaries, indicating that heightened uncertainty contributes to the difficulties that dyslexic children experience in word decoding. The diffusion modeling also contributed to the growing evidence that dyslexic children may experience difficulties with auditory language processing [33]. Although dyslexic children were observed to be equally fast in recognizing auditorily presented words and pseudowords, they were substantially less accurate than the typical readers. More specifically, analogous to the results from the visual lexical decision task, the problems experienced by dyslexic children in performing the auditory version of this task resulted primarily from delays in the evaluation of word characteristics. Unlike visual lexical decisions, however, auditory word decoding does not necessarily rely on orthographic processing, whereas phonological processing is essential. In contrast to visual word decoding, dyslexics’ word decoding in the auditory domain is not hampered by elevated certainty criteria, indicating that uncertainty does not play a role. Importantly, the pattern of findings indicates that the processes of evaluating word characteristics are impaired in both visual word decoding, which is generally agreed to reflect the dyslexic deficit, and in auditory word decoding, in which orthographic influences are diminished. This pattern provides support for the assumption that the core problem of dyslexia resides in the word characteristics essential for both visual and auditory word decoding, that is, in phonological processes [29], p. 1351.

### 3.3. Event-Related Brain Potential Analysis

We examined different components of event-related brain potentials elicited during the early processing of orthographic stimuli. The first component, labeled P1, has a positive polarity and peaks around 100 to 150 ms after stimulus onset. P1 is observed over the posterior occipital brain region. This component is typically associated with low-level features of word processing, such as length and typicality. The second component, labeled N170, has a negative polarity and a peak latency of around 200 ms. N170 is usually observed at parietal-occipital recording sites. N170 has been interpreted to reflect visual expertise and orthographic processing. This component has been demonstrated to arise from the visual word form area [34]. A later positivity, labeled P2, with peak latencies around 300 ms and a temporal topography, has been associated with both phonological and semantic aspects of stimuli used in visual word experiments.

We focused primarily on N170 because this component of the brain potential is found to be sensitive to visual processing and, in particular, to lexical processing [35,36]. Thus, larger N170 amplitudes have been observed for words compared to strings of symbols, shapes, or dots [35]. Our study comprised 19 children with dyslexia (8.97 ± 0.39 years old) and 20 typical readers (8.78 ± 0.35 years old). Bi-syllabic Dutch words or symbol strings derived by converting the word stimuli into a special font were presented in a word reading task. The words or symbol strings were presented during 700 ms and followed by a 1350 ms inter-stimulus interval. Participants were instructed to respond when they detected a target (i.e., when a word or symbol string was immediately followed by itself). Figure 3 presents an example of a trial.

The results that emerged from this study showed a reduced N170 amplitude to words over the left relative to the right hemisphere. This hemispheric difference was not observed for symbol strings and was absent in dyslexic readers. Although this finding is seemingly in contrast with previous research reporting a larger N170 to words than symbol strings [37], it can be interpreted with reference to the visual word form area. The reduced N170 amplitude over the left hemisphere in typical readers may suggest facilitated lexical access. Thus, typical readers might have benefitted from a whole-word level specialization for the current (all familiar) words. Such a beneficial effect might be reduced in children with dyslexia. This is consistent with their underperformance on word reading tests that was observed in this study. Moreover, on the experimental task, children were also less accurate in detecting word repeats than typical readers, which may suggest a specialization deficit at the whole-word level. Collectively, the results from this study suggest that N170 amplitude may provide a sensitive tool to assess visual word processing involved in reading fluency.

### 3.4. Graph Analysis of EEG

The final studies to be reviewed in this section are concerned with a graph analysis of the resting- and task-related (EEG). Graph analysis permits the modeling of the organization of whole-brain functional connectivity [38]. Basically, a ‘graph’ represents a network consisting of a set of nodes (‘vertices’) and the connections between them (‘edges’). Various graph measures permit the characterization of graph topologies in terms of the efficiency and organization of the information flow and the balance between segregation and integration of the network [39,40]. Performing the graph analysis of the resting EEG, we used the Phase-Lag Index (PLI) to assess functional connectivity between all pairs of electrodes. The PLI measures phase synchronization, and it is designed to reduce the effect of volume conduction [41]. We then derived a graph using the minimum spanning tree (MST) approach advocated by Stam et al. [42]. A tree is a loop-less sub-graph derived from a weighted connectivity matrix with a fixed number of nodes and edges, which minimizes bias when comparing between groups or experimental conditions [43]. Nodes with only one link in a tree are referred to as ‘leaf’ nodes (or leaves), and the number of those nodes in a tree is the leaf number, which is used together with other metrics to describe the tree configuration [44].

We collected 2 min eyes-closed resting EEG from 64 channels preceding an experimental session [45]. The EEG data were from 29 children with dyslexia (8.96 ± 0.40 years) and 15 typical readers (8.75 ± 0.31 years). Artifact-free epochs were selected from the 2 min EEG and re-referenced to the average of all channels before performing spectral power analysis and the subsequent connectivity and MST analysis. The analytic steps are presented in Figure 4, and measures that can be derived from MST analysis are presented in Table 2. The MST analysis revealed significant group differences in the theta band of the EEG (4–8 Hz). The group differences pertain to the leaf fraction measure, indicating that the integration of information within the brain network was significantly lower in children with dyslexia relative to the typical readers. Two additional MST metrics, i.e., diameter and eccentricity in the theta band, also revealed differences in network communication efficiency between the groups. It should be noted that theta oscillations play an important integrative role in the organization of brain activity [46]. A recent review by Cainelli et al. [47] indicates that resting EEG in children with dyslexia differs from typical readers in several frequency bands, notably in alpha and theta frequencies. These authors suggest that a theta deficit may alter the modulation of syllable coding and multisensory processing, implicating auditory-visual integration. In line with this suggestion, Archer et al. [48] argued that the inadequate temporal sampling of speech-sounds in people with dyslexia results in poor theta oscillatory entrainment in the auditory domain and, thus, a phonological processing deficit, which hinders reading ability. 

MST results from a follow-up study [49], using data obtained from 28 dyslexics (23.14 ± 2.18 years old) and 36 typically reading adults (22.22 ± 2.52 years old), revealed group differences in the EEG alpha band (8–13 Hz) that are depicted in Figure 5. More specifically, compared to typical readers, dyslexics showed higher degree (i.e., number of neighbors for a given node) and kappa (i.e., broadness of the degree distribution). It should be noted that the pattern of adult findings deviates from the network results observed for children with dyslexia. In children, the primary differences between groups were found in the theta band, while group differences in adults occurred in the alpha band. This apparent discrepancy might be related to maturation. Theta in young children may be functionally comparable to alpha in adults. Studies suggest a shift towards higher frequencies with maturation [50] and shared topographies between higher frequencies in adults and lower frequencies in children [51]. Interestingly, we observed that MST metrics were correlated with age, although the age span under consideration was quite limited. With advancing age, the relevant MST metrics of the participants with dyslexia approached those observed for the typical readers. This pattern could be taken to suggest that a proportion of the dyslexics exhibit a protracted maturational course of brain network topology. This would be in line with suggestions that dyslexic children experience a developmental delay [52] or are lagging behind in the acquisition of reading-related skills [53].

Finally, we conducted a graph analysis of 64 channel EEG while 24 dyslexic participants (22.99 ± 2.29 years) and 31 adult typical readers (22.27 ± 2.53) performed a letter-speech sound mapping task [54]. On each trial, a visual symbol and a phoneme were presented simultaneously, and the participant had to decide whether the two stimuli belonged to each other by pressing one of two response buttons. The performance results showed that accuracy increased with time-on-task but did not differ between groups, and the speed of responding was only marginally slower for individuals with dyslexia. The analysis of the EEG indicated a task-related decrease in functional connectivity (PLI values) in dyslexics compared to controls but no group differences during rest. Here, it should be noted that the pertinent literature revealed inconsistent findings, possibly due to the variety of tasks and methods used. The analysis of MST metrics revealed significant rest vs. task changes, indicating a less integrated network topology during tasks compared to rest. Importantly, relative to controls, a lower tree hierarchy in the theta band was found for participants with dyslexia during task performance. This group difference was absent during rest. A lower tree hierarchy is suggestive of a sub-optimal balance between efficient communication and the prevention of an overload of hub nodes in the network. In addition, relative to controls, a higher kappa in the alpha band was observed for dyslexics during tasks compared to rest. High kappa represents the presence of high-degree nodes, which facilitates synchronization of the tree but may increase the vulnerability of the network if a hub is damaged. We failed to observe a reliable relation between the group differences in MST metric and task performance. It should be noted, however, that in order to increase comparability between rest and task, we used 4 min EEG epochs for data analysis. Obviously, adopting such a wide analysis angle ignores differences between cognitive processing during trials and pause during the interval between trials. Future studies using an event-related analytic scheme should address this issue.

### 3.5. Interim Summary and Conclusion

What did we learn from our studies aimed at obtaining a deeper understanding of dyslexia? First, our findings on letter-speech sound mapping and word decoding strengthened our belief that phonological deficits belong to the core of reading difficulties. Our studies using artificial script convincingly demonstrated that children with dyslexia are characterized by disrupted letter-speech sound learning. One of the advantages of using an artificial script is that it rules out a priori between-group differences in familiarity with the stimuli used to assess letter–speech sound mapping competence. Furthermore, in order to rule out the potential advantage of typical readers having more reading experience, we conducted a study on preliterate children at familial risk for dyslexia and their typical risk peers. Similar to the other studies, the artificial script task discriminated between groups, and, even more importantly, the performance on this task predicted later word reading. Together, our findings demonstrate a pronounced letter–sound mapping deficit in children with dyslexia. We augmented our performance findings by using information that can be extracted from the EEG. The graph analysis of the EEG associated with letter-speech sound mapping showed differences in network topology between individuals with dyslexia and controls during the learning of letter–speech sound mappings, but these EEG differences were not related to performance. Accordingly, future research is needed before attaching importance to these findings. The results obtained for the resting EEG indicated less efficient brain networks in individuals with dyslexia, relative to controls, suggestive of reduced functional specialization of the dyslexic brain compared to typical brains. This suggestion is in line with our brain potential findings indicating the lateralization of N170 amplitude in typical readers that was absent for children with dyslexia when performing a word decoding task. Finally, diffusion modeling of word decoding performance revealed a two-fold deficit in dyslexic children. They experience difficulties in extracting phonological information from the visual stimulus, and, in addition, they adopt more caution before emitting a response. The former finding underscores the alleged phonological deficit in dyslexia, while the latter finding points to enhanced uncertainty when presented with the need to process print.

## 4. Assessment

We performed several studies aimed at the assessment of dyslexia. In this section, three studies that used widely different methodologies will be reviewed. The first study, reported by Aravena et al. [55], took advantage of artificial script learning in the assessment of dyslexia. The second study, reported by Rezvani et al. [56], used graph metrics derived from resting state EEG data to examine whether machine learning could be employed in the classification of individuals with dyslexia. The third study, conducted by Verwimp et al. [57], performed a network analysis to assess the multidimensional character of dyslexia. Network analysis may have considerable implications for the understanding and assessment of dyslexia and in designing intervention procedures.

### 4.1. Assessment Using Artificial Script

Typically, dyslexia is associated with two phonology-related problems—difficulties in phonological awareness (PA) and difficulties in rapid automatized naming (RAN) [58]. PA refers to the ability to identify and manipulate speech sounds and is usually assessed by using tasks in which speech sounds have to be segmented, blended, replaced, or deleted. PA is critical in reading acquisition [59], and PA deficits have been found to be strongly associated with difficulties in reading and spelling [24]. RAN refers to the speeded naming of a series of familiar visual items, such as alphanumeric stimuli, colors, or objects. Difficulties in rapid naming are strongly related to fluent reading problems [60]. In Aravena et al. [55], we used an augmented version of the artificial script learning task to examine whether measures derived from this task may differentiate between individuals with dyslexia and typical readers over and above PA and RAN indices. The augmented version included both accuracy and speed of letter–speech sound mapping. The inclusion of the speed measure is of interest for two reasons. Methodologically, adding reaction times is useful when accuracy measures may reach the ceiling. Theoretically, reaction time might be instructive as previous neuroimaging studies demonstrated that the processing of letter–speech sound correspondences is primarily manifested in the time course of the activation of neural units involved in audiovisual integration as well as in the latencies of overt responses [6,12]. We used a phoneme deletion task to assess PA and a task requiring the naming of letters and digits aloud from a computer screen to assess RAN. Further, word reading was measured using a computerized task including high-frequency words, low-frequency words, and pseudo-words. Word reading in the artificial script was also measured (number of correct words read per second). Finally, spelling recognition was measured by presenting a word over headphones with simultaneous presentation of the word on a computer screen with a letter or letter combination missing. The participant had to decide, as quickly and accurately as possible, which of four letters or letter combinations represented the missing part of the word. The latter two tasks were taken from the task battery constructed by Blomert and Vaessen [11]. We composed a group of 72 dyslexic children using conventional diagnostic procedures and a group of 46 children with average or above reading and spelling skills (age range between 7.33 to 11.08 years). The children received 20 min training on the artificial script learning task.

The results indicated that artificial scripts can be used to differentiate groups. Typical readers performed more quickly and accurately on the artificial script learning task compared to children with dyslexia. Moreover, the number of words read in the artificial script was higher in typical readers than in children with dyslexia. This pattern of findings replicates our previous findings [16] and is in accord with notions that deficits in letter–speed sound mapping are a critical factor in dyslexia. Logistic regression analyses indicated that both accuracy and speed on the artificial script learning task made a significant contribution to group membership. Classification based on these two measures was accurate at 68.6%. Further, correlation analyses showed that the number of words in the artificial script was significantly associated with the PA and RAN measures, and the speed of the artificial script learning task correlated with RAN, even controlling for individual differences in baseline speed. The speed of the artificial script reading task correlated moderately with the speed of word reading and significantly with the speed of spelling recognition. The accuracy of reading the artificial script was associated with the accuracy of word reading and word decoding. Collectively, these findings indicate that the measures derived from a brief 20 min practice on the artificial-script task, together with those from the subsequent artificial script reading, produce a coherent pattern of associations with conventional measures of phonological skills that are used to characterize dyslexia.

It was observed that, with regard to the accuracy of word reading, the number of words read per second in the artificial script contributed more than half of the variance after controlling for age-related variance. Considering the speed of word reading, RAN contributed most of the variance (54%), followed by the number of words read per second in the artificial script. Accuracy of spelling was best predicted by accuracy on the artificial script-learning task. For the speed of spelling, RAN and the speed on the artificial script learning task contributed an approximately equal amount (39% and 36%, respectively). In brief, results that emerged from this study demonstrate that brief practice on the artificial script learning task differentiates between typical readers and children with dyslexia. The measures derived from this training relate to PA and RAN and make a substantial and unique contribution to predicting individual differences in reading and spelling ability when compared with these conventional predictors. It should be emphasized that the 20 min practice refers to learning artificial letter–speech sound mappings. This implies that there are no a priori differences in exposure to the stimuli at the start of the assessment. In this respect, the training provides a relatively “pure” assessment as compared with traditional instruments. More specifically, the typical reciprocity between reading development and phonological development is circumvented [55], p. 561. Traditional tests can be used to determine if letter–speech sound correspondences are weak, but they do not tell whether this should be attributed to a predisposition, to reading problems, to differences in exposure, or to a combination of these factors. In contrast, the current learning procedure allows for the detection of a fundamental learning deficit for letter–speech sound associations [61], p. 166.

### 4.2. Assessment Using EEG Graph Metrics

Dyslexia is considered a complex neurobiological disorder characterized by a varied pattern of structural [62] and functional [63,64] abnormalities. Accordingly, it may not come as a surprise that investigators are searching for biomarkers that can be used in the classification and intervention of dyslexia. Quantified EEG is an early methodology used for the detection and classification of attention and learning disorders, including children [65,66]. This approach relies on the automatic acquisition of EEG samples that are used to provide reproducible estimates of the quantitative features of the EEG (e.g., power and coherence of EEG spectra) [67]. More recent developments take advantage of machine learning methods for constructing EEG-based classifiers [68]. Basically, machine learning consists of using an algorithm for the construction of a model based on a sample set of data, called the training set, that can then be applied to data that the model has never seen before [69]. There are several classification techniques used in machine learning. The Support Vector Machine (SVM) belongs to perhaps the most popular technique for detecting biomarkers based on EEG features because of its sound mathematical basis, simplicity, and low computational complexity [70]. The idea behind SVM is the separation between two classes of data (e.g., dyslexics vs. non-dyslexics). The SVM aims to find a hyperplane that distinctly classifies data points in an N-dimensional space where N stands for the number of features characterizing each data point [71,72]. Support vectors refer to data points that are closest to the hyperplane and influence its orientation. The use of these support vectors will increase the margin of the classifier so that the distance from the hyperplane to the nearest data point on each side of it is maximized. The existence of a maximum-margin hyperplane will ensure that future data points can be classified with greater confidence. The maximum-margin hyperplane could be a linear classifier, but in many instances, classification involves solving a non-linear problem. This can be achieved by applying the kernel trick, which entails the transformation of the data in such a manner that they become linearly separable.

In a preliminary study [56], we used machine learning to classify dyslexics vs. non-dyslexics participating in the study reported by Fraga et al. [45]. The input features in this study were the PLI and MST metrics characterizing the global brain network during rest. A total of 1792 (4 frequency bands × 7 network measures × 64 electrodes) were extracted for each child. After FDR correction for multiple comparisons 37 features remained that were fed to the learning machine. Several classifiers were tested, as it is difficult to determine in advance which kernel function suits the date best. Comparisons between classifiers indicated that two kernel functions yielded good classifications—linear and degree-3 polynomial. Polynomial kernel functions do not only consider given input features of data points but also combinations of these. The SVM was compared with a commonly used simpler classifier—the k-nearest neighbor (k-NN) algorithm [70]. K-NN is based on the intuitive idea that data points close to each other resemble each other most. Accordingly, the distance between a specific query point and its neighbors is calculated and used to form decision boundaries. The value k in the k-NN algorithm defines how many neighbors will be checked to determine the classification of the query point. For example, if k = 1, the query point will be assigned to the same class as its closest neighbor. When query points have neighbors from different classes, a majority vote will resolve the issue. Unfortunately, there is no magic way to determine an optimal value for k. A simple solution is to use various values of k and then select the one producing the best outcome. We selected k-values of 3 and 7. Leave-one-out was used for cross-validation. This method entails training the classification model using all observations but one and then applying the resulting model to predict the one remaining observation. This is repeated *n* times, where *n* refers to the number of observations in the original data set. The performance of the model is examined for each prediction, and an overall measure is obtained to indicate its skill. This procedure is used to avoid overfitting and ensures the reliability and generalizability of the results to new data sets. The various steps in our machine learning classification of the dyslexic vs. non-dyslexic participants are depicted in Figure 6. The performance of the SVM and k-NN models is presented in Table 3, using the conventional measures of precision, specificity, sensitivity, and accuracy.

Table 3 shows that both classifiers worked quite well. Actually, the prediction rates based on global EEG network features were substantially higher than the accuracy rates observed in our previous study based on the performance of the artificial script learning task [53]. The table indicates that the SVM classifier did better than the n-NN ones and that the SVM with a linear kernel performed better than the one with a polynomial kernel. This observation adds to the SVM’s popularity in the classification literature using features extracted from neuroimaging [68]. The results that emerged from our EEG-based classification study should be considered preliminary. The classification was performed only within the available data set. In order to test the robustness of our findings, the classifier’s performance should be evaluated using additional data sets. Further, classification was based on the data from twice as many individuals diagnosed with dyslexia as typical readers, while the prevalence rate of dyslexia is about 4%. Moreover, the sample consisted of extreme groups (i.e., readers with average or above reading performance vs. readers with a percentile reading score lower than 10%), while such a gap in reading performance may not exist in the real world. Collectively, these potential caveats may affect the performance of the classifier and its application in natural settings [73,74].

Although our findings should be interpreted with caution, they contribute to the rapidly growing literature showing that machine learning classification of dyslexia based on features extracted from the EEG holds considerable promise—in particular, when classification is based on tasks specifically designed to examine those processes that are thought to be compromised in dyslexia. A recent study focusing on mathematical achievement provides an interesting illustration of such an approach [75]. This study included children who scored low, average, or high on the math section of an achievement test and were asked to perform a numerical comparison task while their EEG was recorded. Graph metrics derived from the task-related EEG were submitted to a machine-learning algorithm, known as Decision Tree, that yielded an appropriate classification accuracy (70 to 80%). This method also allowed for the identification of the electrodes that were involved in the classification. This information could then be used to examine group differences in the brain regions activated during task performance. It was observed, for example, that average and low-achievement children differ in the recruitment of the parietal and frontal cortex, which is consistent with findings suggesting that with the advance of mathematical expertise, brain activation shifts from frontal to parietal regions. It has been suggested that machine learning classification based on task-related EEG may provide a strong tool in the identification and remediation of children with reading problems in school settings [76]. This does not need to be wishful thinking. Consumer-friendly brain-computer interfaces for collecting EEG data equipped with open-source machine-learning tools are in the making [77]. Accordingly, the application of EEG-based classification in schools is rapidly approaching. Only recently, a group of Turkish investigators developed a dyslexia biomarker detection app using a machine learning classifier of 14-channel EEG acquired by cost-effective mobile brain wear that is claimed to be suitable for home and school use [78].

### 4.3. Assessment Using Network Analysis

The third approach towards the assessment of dyslexia is concerned with a network analysis that gained rapid prominence in the study of psychopathology. We started the current paper by referring to the IDA definition of dyslexia. This definition conceives dyslexia in terms of a phonological deficit that can be traced back to a neurobiological mechanism and is manifested by difficulties in fluent and accurate word reading and poor spelling and decoding abilities. Such a definition is akin to the conception of medical disorders in which a set of symptoms (e.g., fever, headache, coughing) result from a bodily ailment (e.g., a viral infection). Dyslexia is different from a medical condition in that it cannot be demonstrated independent of its symptoms, which occurs in medical practice (e.g., taking a blood sample or a throat swab for diagnosing a viral infection). In this regard, dyslexia is taken as a hypothetical construct invoked to explain why certain reading difficulties hang together, and this notion is reinforced by psychometric practice linking reading problems to latent variables.

Network analysis provides an alternative approach for assessing dyslexia similar to its use in psychopathology. Borsboom and his co-workers proposed a network model of psychopathology, assuming that symptoms are not caused by an underlying disorder but are constitutive of the disorder [79]. Network analyses of psychopathology conceive symptoms as part of a causal system. They are mutually interacting elements of a complex network. The disorder does not produce such a network. The disorder coincides with the network that consists of elements (nodes) and the relation between them (edges). The nodes can be virtually anything (persons, airports, but also indicators used in the assessment of dyslexia). Edges can be weighted or unweighted; the latter signifies the magnitude of the connection, while the former merely indicates that two symptoms are connected. The relation between two nodes can be positive or negative. When the association between symptoms is positive, they are mutually reinforcing. When negative, they counteract each other. Further, edges can be directed or undirected. The edges of undirected networks indicate that pairs of symptoms are connected, but the network is agnostic about whether one node influences the other or vice versa. Directed networks consist of edges pointing in the direction of influence or causation. Finally, some nodes in the network are more important than others. Highly central nodes are most important but don’t need to be unique to a disorder. For example, a highly central node may link sub-networks together, thereby producing diagnostic comorbidity. Different centrality measures have been proposed. We consider three of them—node strength, closeness, and betweenness. Strength centrality denotes the number and strength of the edges connected to a certain node. It refers to the probability that this node will be activated by the nodes connected to it. Closeness is a centrality measure that provides an estimate of the average distance of a particular node to all other nodes in the network. The betweenness of a node indicates the number of times a node lies on the shortest path between each other pair in the network.

At this point, it should be acknowledged, however, that it has been argued that the idea of node centrality is not well suited for how psychological variables may relate to each other and that, consequently, one may better focus on the network as a whole or, alternatively, develop other centrality measures of alternative measures of variable importance [80]. In addition, it has been indicated that the role of network analysis in generating causal hypotheses remains a challenge both from a technical point of view [81] and from a philosophy of science perspective [82]. A critical analysis from a psychometric perspective on the classification of psychopathology can be found in Hallquist, Wright, and Molenaar [83].

Our network analysis was performed on the data of 1257 children (age range between 85 and 158 months) who were selected from a larger archival data set containing diagnostic data collected by a nationwide clinical center for learning disabilities in the Netherlands [57]. In total, 28 reading-related variables, including cognitive, demographic, and environmental measures, were used to construct the network that is presented in Figure 7. Each edge in the network represents an undirected relation controlling for all other variables in the network. Stronger relations are represented by thicker edges, positive relations are indicated by blue edges, and negative relations are represented by red edges. Absent edges should not be taken to imply that a variable is not marginally related to reading but indicate that there is no relation given all other variables in the network.

Upon visual inspection of the network, two clusters become immediately apparent; one cluster is associated with reading, and the other with intelligence. In addition, it can be seen that while RAN is particularly involved in fluent reading, PA is important in accurate reading. This observation is in line with the notion that the abilities tapped by RAN and PA follow different time-courses during reading development [84]. PA is related to the controlled and accurate processing of speech sounds and letter–speech sound mapping (i.e., decoding) and working memory. These processes are thought to be especially important during the initial phases of reading acquisition. RAN is primarily involved in the automated convergence of visual and speech information that is important during later phases toward fluent reading [85,86]. It can be seen that the influence of intelligence is mediated by visual perception, suggesting that children with higher visual perceptual abilities tend to be faster in identifying letter-speech sound associations and are more fluent in selecting the correct letter during the spelling task. It cannot be excluded, however, that this relation is mediated by a variable that has not been considered, such as global processing speed. Peter et al. [87], for example, found lower processing speed in children with reading difficulties [31]. The observation of separate clusters associated with intelligence and reading indicates that more intelligent children do not necessarily read better than less intelligent children. This observation argues against the discrepancy criterion used in the identification of problematic readers. In line with Stanovich’s [88] suggestion, our network finding indicates that reading problems should be evaluated without considering potential reading-IQ discrepancies. Finally, network analysis assumes that targeting critical nodes in the network may elicit a cascading increase or decrease in the activation of the nodes connected to the target node. In the next section, it will be illustrated how intervention may alter the reading network.

### 4.4. Interim Summary and Conclusion

Our studies focusing on the assessment of reading difficulties illustrated three widely diverging approaches. The first approach relied on the learning of artificial script. The results showed that this type of assessment contributed to the prediction of individual differences in reading and spelling ability and moderately but significantly to group membership. The use of an artificial script is of special interest since there are no a priori differences in the stimuli that are used in the assessment. In this regard, this approach provides a relatively ‘pure’ instrument compared to conventional assessment tools. The second approach focused on the neural correlates of reading difficulties. For the first time, machine learning was used on graph metrics derived from the EEG. The results showed that this approach was quite successful in the classification of individuals with reading problems vs. typical readers. It should be acknowledged, however, that these findings are preliminary. Replication is mandatory and should be safe guarded against over-sampling and over-fitting. The third approach exploited network analysis, an approach that gained considerable popularity in the study of psychopathology. The results that emerged from this analysis indicated the prominent role of PA and RAN, thereby validating the network approach. Although PA and RAN appeared to be central nodes in the network, they are activated along different pathways and thus suggest that similar reading outcomes may arise from different loci. The analysis pointed to the complexity of the reading network, indicating that reading difficulties cannot be reduced to a single core deficit [88,89]. Moreover, the complexity of the reading network strongly suggests the need for targeted assessment and for developing personalized interventions. 

## 5. Intervention

In this section, we will review four approaches to intervention. First, we will ask the question of whether reading difficulties can be ameliorated by a computerized intervention inspired by the notion that a phonological deficit lies at the heart of the reading, and perhaps spelling, problems that children experience who are diagnosed with dyslexia. Then, we will turn to the question of whether dynamic assessment using artificial script training may contribute to singling out children who are responsive to intervention versus those who are not, or only to a very limited extent. The next question we asked was whether intervention is associated with a simultaneous change in the neural correlates of compromised word decoding in children diagnosed with dyslexia. Finally, we examined whether the beneficial effects of intervention are manifested in the dyslexia network.

### 5.1. Computerized Intervention

In an early study, Tijms and colleagues examined the treatment effects of a computerized intervention program, coined LEXY, both during the course of treatment and during a four-year follow-up to assess whether the beneficial effects of the treatment would survive [89]. The intervention program was developed in line with the results of an abundance of research indicating that a deficit in the phonological system is a key factor in dyslexia. Furthermore, previous studies revealed the beneficial effects of phonology training on reading performance. These studies indicated that promoting phonological awareness is necessary but not sufficient in attaining improvements in reading and spelling [90]. Collectively, these considerations guided the development of the LEXY intervention. Basically, LEXY is a computer-based intervention and focuses on learning to recognize and use the phonological and morphological structure of Dutch words. More specifically, the focus of LEXY is on the language units, the basic rules, and the minimal heuristic knowledge needed to be implemented in a computer program in order to transform a phonic word into a correct orthographical word form. The sound structure of words is the basis of the treatment. The program focuses on the smallest possible but still naturally pronounceable phonological structure, i.e., the syllable. By taking the ‘spoken’ syllable as the unit of processing, the participant finds it easier to identify distinct speech sounds. The usual qwerty keyboard has been redesigned to include four different key categories. One part of the keyboard contains an abstract token for designating speech segments (e.g., vowels, diphthongs). Another part contains keys for concrete sounds (e.g., /g/or/o/). The third part consists of keys related to bound morphemes (e.g., /ge/or/ig/). The last part of the keyboard contains icons for specific operations. Treatment consists of 45 min sessions, once a week and 15 min home practice three times a week. Treatment consisted of six modules: (1) distinguish speech sounds in monosyllabic words, (2) focus on dissociations between phonics and graphical representation, (3) consider polysyllabic words, (4) focus on the morpheme structure of words and on compound words, (5) attention to Dutch verbs, and (6) consideration of loan words. During treatment, a mastery level should be attained for each element, implying that the duration of the treatment varies. Typically, the treatment duration is about one year. The goal of the LEXY intervention is to achieve a functional level of literacy; that is, a level at which the participant can function at school and, later, in their profession and in society at large.

Tijms et al. [89] recruited 100 participants diagnosed with dyslexia (age range between 7 and 41 years at the start of intervention). The effects of the intervention were evaluated for each participant during the course and at the end of the intervention. Separate groups were composed for successive follow-up evaluations (1 to 4 years following intervention). The evaluations focused on three skills: word reading, text reading, and spelling. A remedial index was constructed to provide an estimate of the intervention effect. The index has a range from 0 to 1, where zero stands for a nil effect and one for reading and spelling performance at the norm group level. The intervention results are depicted in Figure 8; the left panel shows the intervention effect on word reading, the middle panel for text reading, and the right panel for spelling. It can be seen that the LEXY intervention has a strong beneficial effect. By the end of the intervention, the distance in performance relative to the norm group level is halved for word reading and is gone for text reading and spelling. Importantly, the beneficial effects even grow during follow-up for word and text reading. For spelling, the intervention effect is seen to decline somewhat, but it continues to stay at an appreciable level above the performance at the start of the intervention. The apparent discrepancy between reading and spelling is explained by referring to differences in experiences. Whereas all participants are required to read in order to successfully navigate their environment (school, profession, society), spelling is far less wanted and, thus, practiced. Accordingly, it does not come as a surprise that spelling, being the ‘rule-learning instruction kernel’ of the treatment, shows signs of over-learning at the end of treatment, which is corrected after the treatment. Most likely, the partial relapse after the termination of the intervention should be considered a one-time-only event and not a steady decline [1], p. 134. Overall, the results that emerged from this intervention study indicated that remediation focusing on the phonic structure of words provides a promising avenue for tackling the reading and spelling difficulties of individuals diagnosed with dyslexia [91,92,93,94].

### 5.2. Artificial Script in Intervention

We exploited the artificial script learning task employed in Aravena et al. [16] as a dynamic assessment instrument for predicting the response to the LEXY intervention of dyslexia used in Tijms et al. [89]. A central issue in the intervention of dyslexia is concerned with the question of when intervention should be offered, which intervention would be appropriate, and how much intervention is needed for remediation of the reading difficulties. One way of addressing this question is educational screening to detect individuals with reading problems as early as possible, perhaps even before formal reading instruction. Such screening may also provide information with respect to the kind and intensity of intervention that is needed. Screening typically involves the administration of static achievement tests and may run quite easily into prognostic difficulties with the risks of under- or overidentification [95]. The growing dissatisfaction with conventional procedures for assessing learning disorders led the IDEIA (Individuals with Disabilities Education Improvement Act, 2004) to permit the use of alternative methods for identifying children in need of intervention (https://sites.ed.gov/idea/, accessed on 27 December 2023). One such method is considering the child’s response to intervention or instruction (RTI). This procedure may imply examining the individual’s performance during regular reading instruction in the classroom and considering reading progression over time. The examination of RTI has several disadvantages. It is time-consuming, as performance should be tracked across a relatively long period before reading problems become clearly manifest. In addition, RTI is associated with statistical problems in determining cut-offs for separating non-responders from responders. Dynamic assessment (DA) is a third approach for determining the timing, type, and dosage of intervention vis-à-vis the potential reading capabilities of the individual [96]. In contrast to static assessment targeting skills that are current during the time of testing, the focus of DA is on prospective inferences about the individual’s capacity to develop a certain skill (e.g., become a fluent and accurate reader with appropriate instruction and contextual factors). Grigorenko [97] argued that RTI and DA may be sides of the same coin. Although originating from different contexts, these two approaches have a common focus—skill development and rate of learning. Grigorenko [97] points to literature in which the notions underlying RTI and DA have been used interchangeably and cites literature in which the two approaches have been blended [98]. This is in line with the approach adopted by Aravena et al. [99].

The participants in our study were 55 children diagnosed with dyslexia (age range between 7.33 and 11.08 years) [99]. The central question addressed in this study was whether the indices derived from the artificial script learning task developed by Aravena et al. [16] could be used to predict the responsivity to the dyslexia intervention program LEXY designed by Tijms and colleagues [89]. Children took the artificial-script learning task and a subsequent reading task of highly frequent Dutch words written in the artificial script about four months prior to starting the intervention. Typically, LEXY intervention takes about 48 to 68 sessions, with one session a week, but in our study, post-tests were administered at 39 weeks. Conventional measures for the assessment of dyslexia included phonological awareness (PA), rapid automatized naming (RAN), word reading, and spelling. The analysis of intervention effects showed that going from pre- to post-test, children made considerable gain—more progress than the reading gains made by their norm group peers. Despite the improvements, the children still lagged behind norm scores. This observation is not surprising given that post-tests were administered when intervention was still in progress—at 39 weeks. Importantly, in contrast to PA and RAN, the speed measure derived from artificial script learning made a significant contribution to explaining variance in response to an intervention targeting reading accuracy and speed. Further, the number of words read per second within the artificial orthography accounted for another significant portion of the variance in reading accuracy post-test. This pattern of findings contributes to previous evidence that DA is useful in predicting children’s literacy abilities [100], reading development, reading deficits [101], and response to intervention [102]. Our results indicate that a dynamic approach to assessment provides a viable avenue to predict responsiveness to intervention, even for the most reading-disabled. From a clinical point of view, early identification of potential non-responders is valuable because it may assist practitioners in adapting their educational strategies at an initial point [99], p. 213.

### 5.3. Neural Correlates Associated with Intervention

In Fraga González et al. [103] we linked two previous studies, Aravena et al. [16] focusing on letter-speech sound mapping, and Fraga González et al. [35], examining neural correlates of word decoding in children with dyslexia and typical readers. The latter study showed that the amplitude and hemispheric laterality of the N170 brain potential elicited during word decoding differentiated between individuals with dyslexia and typical readers. The question asked in Fraga González et al. [103] is whether the potentially beneficial effects of an intensive letter–speech sound mapping will be manifested by a trend towards normalization of the N170 brain potential pattern. Participants were 18 children diagnosed with dyslexia (9.05 ± 0.46 years old) and 20 typical readers (8.78 ± 0.35 years old). They received an intensive tutor and computer-assisted training program focusing on letter–speech sound mapping. This type of training was selected as previous research indicates that difficulties in the automatization of letter–speech sound correspondences are a key factor in dyslexia [16]. Further, this specific training was demonstrated to produce substantial gains in reading and spelling in a previous intervention study [104]. Letter–speech sound mappings are explicitly trained (first regular and subsequently irregular correspondences), aiming at a step-by-step accurate mastery of the learned associations. The tutor explains the letter–speech sound correspondences to the participant by presenting phonemes both in isolation and within the context of a (visual) word. Then, the child has to identify the phoneme type (e.g., ‘long vowel’), syllable type (e.g., ‘stressed syllable’), and operating rule (e.g., ‘delete a phoneme if the terminal phonic element of a syllable belongs to a certain category’), both orally and by pressing the corresponding buttons in the touch screen. A typical example of an exercise is presented in Figure 9. The child is asked to pronounce the word sound by sound, and in the end the whole word guided by the time constraints of the graphemic presentation rate. Training is adjusted to the individual acquisition rate by adapting time constraints to the level of the child’s performance. The training consisted of 34 sessions of approximately 45 min with two sessions a week.

The EEG was recorded from 64 sites, but the analysis focused on parietal and occipital sites, given the scalp distribution of the N170 brain potential. Children performed a word decoding task using letter and symbol strings in separate trial blocks. The word strings consisted of short (4 or 5 letters) or long (6 or 7 letters) bi-syllabic words that were selected using estimates of age of acquisition. The symbol strings were constructed by converting the same words into a special font. The child’s task was to press a button on strings that immediately repeated the previous one. Children received eight blocks consisting of 44 trials, including four target trials. Trials were the same as in [18] (see Figure 3).

As anticipated on the basis of our previous findings, the training in letter–speech sound mappings exerted substantial effects on conventional measures of reading, with the largest gains for fluency and, to a lesser extent, accuracy. In line with our previous study [35], we observed that, pre-test, the amplitude of the N170 to words discriminated between typical readers and children with dyslexia. For the former, the N170 amplitude was smaller over the left compared to the right hemisphere, whereas this hemispheric difference was absent for the latter. N170 amplitude to symbols was overall smaller than the brain's potential response to words and did not show the lateralized pattern. A detailed analysis of the post-test results revealed considerable individual differences in reading gain. In Figure 10, it can be seen that individual differences in training-induced reading fluency are systematically related to N170 amplitude. A larger gain is associated with a greater reduction in N170 amplitude. A final analysis took N170 amplitude at pre-test to assess whether this brain potential could be used to predict the individual’s response to the training. A median split was performed to separate responders from poor responders. Discriminant analysis indicated that 78% of the responders and 68% of the poor responders could be classified accurately by taking the pre-test N170 amplitude as a predictor.

The pattern of findings that emerged from this study makes a couple of important points. First, the results point to the robustness of the finding that N170 amplitude associated with word decoding is lateralized with reduced amplitudes over the left hemisphere relative to the left in typical but not dyslexic readers. The relatively reduced N170 amplitude over the left hemisphere in typical readers may suggest that they need to spend less effort in the decoding of words than children with dyslexia. In contrast to typical readers, dyslexic children may have to rely more than typical readers on the orthographic features of the words [105]. Secondly, and consistently with a previous study [104], the results demonstrated the beneficial effect of intensive training in letter–speech sound mapping on reading fluency. It should be acknowledged that, in spite of a substantial improvement, the training did not bring children with dyslexia to the proficiency level of typical readers. Previously, we observed that intervention exercises its beneficial effect initially on reading accuracy, with reading fluency lagging behind. This observation was interpreted to suggest that training allows for the development of explicit and systematic word decoding skills that then drive self-regulated mechanisms to bootstrap fluent reading [93]. It would be of interest to assess whether N170 amplitude associated with word decoding will track the protracted course of reading fluency gains. Third, and most important, the pre- vs. post-treatment comparison demonstrated a normalization of the N170 amplitude pattern for responders that was absent for non-responders. Incidentally, the change in the N170 amplitude pattern was not observed when considering the whole sample. In addition, it should be noted that the apparent normalization did not include the lateralization in N170 amplitude seen in typical readers. In this regard, it must be concluded that abnormalities continue to exist following training. It cannot be excluded, however, that, in line with the findings regarding reading fluency, complete normalization of N170 amplitude takes even more time than invested in the present study. Finally, the results showed that N170 amplitude at pre-test discriminated responders vs non-responders to intervention while the reading measures failed to discriminate. This finding extends previous findings indicating that the activation of brain areas for reading is an important factor to consider in designing and providing intervention strategies for individuals who struggle with reading [106]. Together, the pattern of results provides further support for the relation between N170 amplitude and reading expertise and its potential use in the assessment and remediation of dyslexia.

### 5.4. Network Analysis and Intervention

A final illustration pertaining to intervention is from the Verwimp et al. [57] network analysis of dyslexia. For a sample of the children participating in this study (806 children aged between 85 and 158 months) who received an intervention similar to the one in Fraga González et al. [103]. The reading network was re-analyzed using intervention progress as a moderator variable. The resulting network is depicted in Figure 11. It can be seen that an appreciable number of associations disappeared compared to the overall network (see Figure 8). More specifically, the cluster comprising reading accuracy and fluency variables remained, but L–SS and spelling fluency were no longer connected to the other variables in the network. Similarly, the intelligence-related measures and reading measures are now disconnected as the associations with the variables that funneled this relationship in the general network disappeared (visuo-constructional abilities and visual perception). The isolated relation of PA with the reading cluster is consistent with studies indicating that interventions specifically targeting phonological awareness seem to offer little solace. At this point, it should be noted that the current network illustration is only scratching the surface. Network analysis allows for the assessment of dedicated interventions targeting major nodes whose activation will spread across the network. A concise illustration of the usefulness of network analysis in examining treatment effects is offered by Blanken et al. [107]. These authors used network analysis to assess the potentially beneficial effect of cognitive-behavioral therapy on concurrent insomnia–depression problems. The participants received treatment for five successive weeks, and symptoms of insomnia and depression were recorded for 10 weeks, 2 weeks before treatment, during treatment, and three weeks after treatment. Each week, a network was estimated that included all symptoms and treatment as a separate node. In this vein, the network analysis allowed for an examination of the time course of the differential patterning of treatment effects. In principle, a similar approach could be adopted when intervening in dyslexia. Moreover, such an approach can be personalized by infusing network analysis with time-series data, as has been demonstrated by Epskamp and co-workers [108]. A further development in network analysis is the connection with controllability statistics that can be used to probe the effects of interventions along the way and allows for a judicial selection of targets depending on the course of treatment effects [109]. This development was already foreshadowed by our former colleague Peter Molenaar [110].

### 5.5. Interim Summary and Conclusion

Our intervention-related research addressed four important issues. First and clinically most important, our research focused on developing an intervention that should be doable and effective. The construction of such an intervention was inspired by the notion that the phonics of language are crucial in the remediation of dyslexia, and the components of the intervention were based on linguistic principles. The application of the intervention demonstrated improvements in both reading accuracy and fluency, with accuracy gaining more rapidly than fluency. Importantly, the beneficial effects of the intervention were seen to persist after treatment. Although the reading gains may not reach the level attained by typical readers, the improvements that were obtained were such to allow for a regular school career and/or a reading proficiency meeting professional and societal demands. Secondly, our research demonstrated that brief training in letter–speech sound mappings using artificial script predicts the dyslexic’s response to intervention. This observation contributes to the advantages of the use of dynamic assessment in determining, on a case-by-case basis, whether the administration of a specific treatment is meaningful. A responsible estimate of intervention success is cost-effective and, with an eye to the client, an ethical precondition. Thirdly, we asked whether the N170 amplitude response to word decoding mirrors at the neural level the beneficial effects of training in letter–speech sound mapping. The results were moderately positive in showing a trend toward the brain’s potential pattern of typical readers after intervention. The amplitude of N170 was reduced relative to the pre-test, suggesting that the intervention had a beneficial effect on the amount of effort needed in word decoding. However, the apparent normalization was not complete; that is, lateralization towards the left hemisphere. The normalizing trend was only seen in responders supporting the validity of our N170 amplitude findings. Finally, we showed the results of a network analysis of dyslexia. An interesting feature of network analysis is that the nodes in the network are constitutive of the disorder under consideration, which may have important implications for our thoughts on dyslexia, understood as a disability that has a neurobiological origin that manifests itself primarily in phonological language deficits. The revolutionary idea of network analysis is turning our traditional conception of what makes a disorder upside down. Finally, we illustrated how the dyslexia network may change as a result of intervention. This illustration must be considered highly preliminary but was included to make the point that network analysis offers interesting avenues for designing tailor-made interventions and offering the opportunity for online control and adjustments.

## 6. Conclusions

We reviewed a series of our dyslexia studies, departing from the notion that phonology is key to understanding reading difficulties and using a variety of methods. In several studies, we focused on letter–speech sound mappings using an artificial script in order to exclude prior differences in reading experience. The results that emerged from these studies demonstrated that even a brief training session using a gamified tool contributes to the prediction of future reading difficulties and responsiveness to intervention in children with familial risk. In addition, the use of this artificial script tool benefits classification and improves word reading. The results of the brain potential and EEG studies provided further support for the neural basis of word decoding problems in individuals diagnosed with dyslexia. In addition, these studies revealed that the beneficial effects of a phonology-inspired intervention are associated with a trend toward normalization of the brain potential associated with word decoding. Graph analysis of the EEG indicated that the global brain network in individuals with reading difficulties is less efficient compared to controls. A further study demonstrated the usefulness of applying machine learning methodology in the classification of dyslexics based on the metrics derived from graph analysis. Several studies were devoted to the assessment of the potentially beneficial effects of the computerized LEXY intervention program that is based on phonological and linguistic principles and proceeds in a stepwise fashion. These studies indicated that the intervention improves not only reading accuracy but also the fluency of reading that, frequently, is more difficult to remedy. Moreover, longitudinal analysis indicated that the beneficial effects persist over several years post-intervention. Finally, two modeling studies were presented, one using diffusion modeling and the other conceiving dyslexia in terms of a network consisting of constitutive indicators. The former study demonstrated that the problem encountered by dyslexics in word decoding is two-fold—a slowness in the accrual of phonological information from the script, together with greater caution in executing decisions based on the acquired information. The latter study presented the first application of network analysis to dyslexia and the sensitivity of this network to intervention. The underlying idea of network analysis is that dyslexia is not conceived in terms of a collection of indicators that hang together because of some latent construct. The indicators themselves are constitutive and might provide targets for intervening in the dyslexia network.

We started this paper by referring to the IDA definition of dyslexia, which considers dyslexia a disorder with a neurobiological origin. This notion of dyslexia is not uniformly shared in the field. It has been argued that reading problems have been unjustly biologized. Lopes [111], for example, suggests that the now dominant neurobiological perspective on dyslexia can be traced back to its ‘founding father’, Rudolf Berlin, who held that difficulties in reading must arise from some anatomical abnormality that had yet to be proven. This possibility is strongly denied by Lopes [111], who argues against the categorization of reading difficulties and maintains that there is not a single shred of evidence indicating that those at the lower end of the reading continuum suffer from a neurological disease. Then he asks what would happen should piano playing be an obligatory part of the school curriculum:


*“It is interesting to speculate what would happen if everyone was expected to learn how to play the piano. Perhaps then schools would be full of children who failed because of a piano disability ‘originating from congenital brain’ and ‘phonological abnormalities” [111], p. 226.*


More recently, Protopapas and Parila [112] used a similar analogy to argue against dyslexia conceived as a disorder arising from a neurobiological origin.


*“Little Johnny was in distress. He had been taking singing lessons for a few years already, but obviously, this wasn’t working for him. Every time he tried to sing he could see others cringe…His highly musical family was gravely concerned: Everyone else was an accomplished singer or on the way to becoming one, but for the life of him, Johnny just couldn’t sing in tune. He was taken to a specialist…[who]…said Johnny should not worry because it was not his fault. His brain was just miswired…Johnny was diagnosed with a disorder and was prescribed intervention to tackle his disability…Had it not been for the singing lessons, nobody would have ever come up with the idea that anything might be wrong with him” [112], p. 1.*


The basic message that can be derived from these analogies is that dyslexia should not be considered a neurodevelopmental disorder but rather in terms of a lack of skill. Indeed, Protopapas and Parila [112] defined dyslexia in terms of a skill:


*“[Dyslexia is] a persistent and unexpected difficulty in developing age- and experience-appropriate word reading skills” (p. 3).*


It is of interest to follow the reasoning that led them to adopt a skill-based definition of dyslexia. First, they argue that the alleged anatomical underpinnings of the disorder are hard to find. They referred to a meta-analysis conducted by Ramus et al. [113] that revealed that most studies examining the neural substrate of dyslexia are too small to derive meaningful conclusions. Further, studies are typically fraud with methodological issues. It should be noted, however, that this argument against dyslexia as a disorder is not principled. Methodologies can be improved, and sample sizes increased. Recently, methods have been developed to determine the optimal sample size when collecting data is in progress [114].

Secondly, Protopapas and Parila [112] argue that the available evidence is based on group comparisons and is only correlational, not causal. The pattern of activation observed in a group of good readers and the deviation from this pattern observed in a group of poor readers does not necessarily imply that any of the poor readers actually exhibit the deviating pattern of activation. Again, this is not a principled argument against dyslexia as a neurobiological disorder. Observing group differences can be augmented by an analysis of individual members to assess whether they actually display the average pattern. Methods are now becoming rapidly available for addressing this and similar questions in the context of small samples and N = 1 studies [114].

The criticism that most of the available evidence is correlational, not causally constitutive, is not convincing. Several meanings of “causality” are in the philosophy of science literature. For example, Woodward’s [115] influential interventionist theory of causation entails that causal relations should be evidenced by the changes that are brought about by surgical intervention. That is, the intervention should affect only the target variable or the path including this variable, and the observed change cannot be produced otherwise. Focused lesion studies [116] targeted brain stimulation studies [117] or conceptually guided intervention studies [104,118] provide such causal evidence.

Thirdly, Protopapas and Parila [112] argue that reading is a skill and, just as any other skill, reading is supported by the brain. Brain differences between typical and poor readers may emerge by exercising this skill to various degrees.

Therefore, the alleged brain differences between typical and poor readers should be demonstrated in preliterate children before the start of formal reading instruction. That is exactly what has been performed in the structural MRI studies included in the meta-analytic review by Vandermosten [119]. This review indicates that differences between at-risk children and controls are consistently observed for the left temporo-parietal and, to a lesser extent, for the occipito-temporal brain regions. The authors suggest insufficient functional and structural connectivity among these regions may result in behavioral deficits in children with dyslexia. This suggestion seems to mesh well with our findings, indicating functional differences in the same areas [35] and aberrant brain networks in individuals with dyslexia [49].

Finally, Protopapas and Parila [112] argue that demonstrating brain differences between at-risk children and controls is not sufficient for conceptualizing dyslexia as a disorder. The brain differences that might be observed should not just be ‘different’ but ‘atypical’. In this regard, dyslexia is definitely not on par with established neurodevelopmental disorders, such as Fragile X or Williams syndrome. The behavioral deficits associated with such disorders are pronounced and widespread in contrast to the specific reading deficit(s) seen in poor readers. Moreover, prevalence rates of these disorders are estimated to be 1:7500 or less, which is a fraction of the 10% proportion of the population that is considered to be dyslexic. The issue of ‘different’ vs. ‘atypical’ basically amounts to a discussion of whether dyslexia should be seen as a categorical vs. dimensional disorder. This issue is not resolved in the dyslexia literature. Some call for the need for a taxonomy, a principled classification system that should link behavioral manifestations to some underlying mechanism [120,121], while others argue for a dimensional approach without a clear cutting point [14].

From a psychometric point of view, the issue is to determine whether the latent nature of manifest variables is category- or dimension-like. This is not as simple as it might appear; categories might be dimension-like and dimensions category-like, and hybrid variations are common [122,123]. There is a strong movement arguing that psychopathological classification based on expert consensus is obsolete and should be replaced, or at least informed, by a system emerging from a progressive accumulation and integration of research findings across various levels of analysis, from genes to self-report, and across various domains or constructs [124]. Astle et al. [125] adopted this research domain criteria (RDoC) framework for reconceptualizing neurodevelopmental disorders and, more specifically, Talcott [126] advocated the use of this framework for obtaining an understanding of dyslexia that is based on the judicious analysis of multi-factorial risk factors (genetic, neurologic, cognitive, environmental). This approach is considered to assess time-varying neurobiological developmental disorders vis-à-vis intra-individual variability along the pertinent dimensions.

The RDoC framework represents the growing unease with conventional diagnostic systems based on broad heterogeneous classes. It is an explicitly agnostic attempt at clarifying how symptomatic expression may arise from disruptions at various levels of analysis in order to allow for a more precise treatment [124]. It should be noted, however, that this framework still relies on constructs, now conceptualized in terms of heterogeneous dimensions, that are somehow linked to multiple response systems (i.e., the units of analysis) operating in neurodevelopment vis-à-vis the environmental context. The network analysis illustrated in this paper represents a further step in this development by assuming that indicators emerging from the analysis of the response systems are causally constitutive without the need to resort to an underlying construct as the common cause of the indicators [127]. Differences in network structure may give rise to individual differences in indicator dynamics. In strongly connected networks, for example, a change in one indicator may result in the full and persistent activation of other indicators, while in weakly connected networks, such a change results only in a limited and temporary activation of other activators. In this regard, the network analysis speaks to the category vs. dimension issue. For individuals characterized with strongly connected networks, the apparent disorder is category-like, while for individuals with weakly connected networks, the disorder is dimension-like. In presenting an illustration of the application of network analysis to the study of dyslexia, we made the first step on what we are convinced is a viable avenue for examining the within-person dynamics of individuals who struggle with reading. The results that will emerge from such an endeavor will allow for designing timelier and person-oriented interventions.

## Figures and Tables

**Figure 1 behavsci-14-00072-f001:**
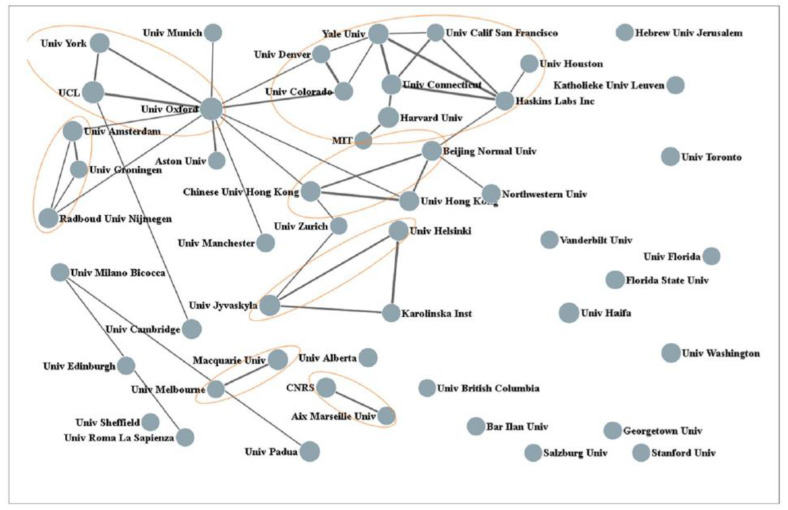
Collaboration Network [5].

**Figure 2 behavsci-14-00072-f002:**
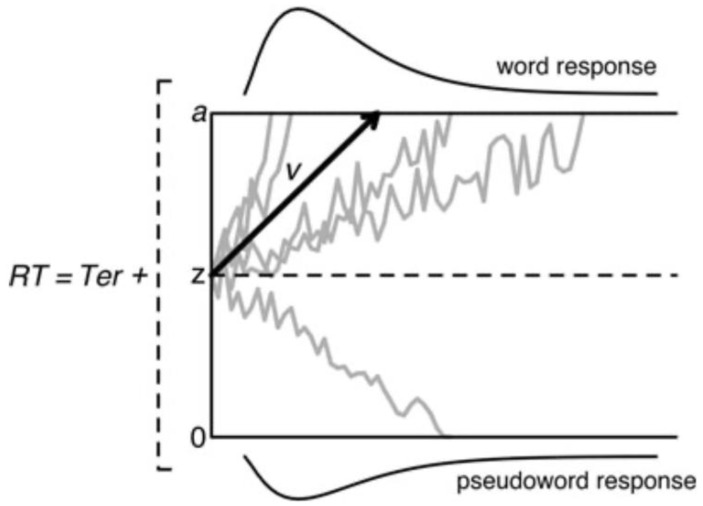
Illustration of the diffusion model in lexical decision. The accumulation of information that drives the decision- making process begins from a starting point (*z*). It continues over time until it reaches either the upper threshold, associated with a ‘word’ response (*a*), or the lower threshold, which is associated with a ‘nonword’ response (0). Each erratic line represents the process of information accumulation for a single stimulus letter string. The drift rate (*v*) is the average rate of information accumulation towards one threshold. The boundary separation (*a*) describes the distance between thresholds and is determined by the amount of information needed before a decision is made. The non-decision component (*Ter*) consists of all the processes other than the decision process that are included in the reaction time (RT).

**Figure 3 behavsci-14-00072-f003:**
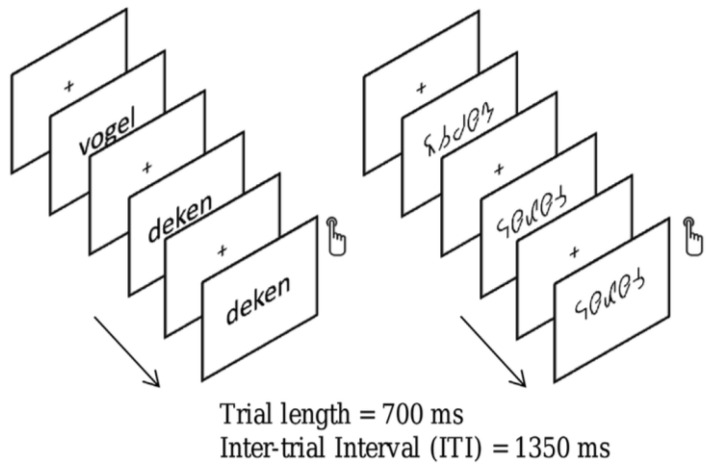
An illustration of the word and symbol strings used in the present study. Children were required to attend to the strings and to depress a button whenever a string was identical to its immediate predecessor. Strings of words and letter-like symbols were presented in a blocked design. A fixation cross was presented in between strings [18].

**Figure 4 behavsci-14-00072-f004:**
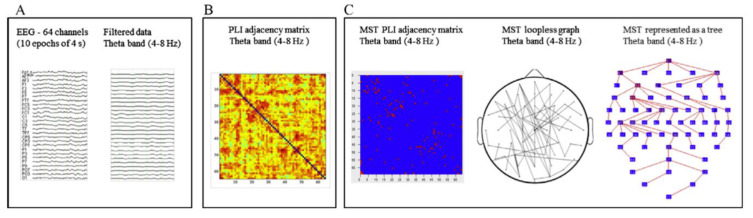
Schematic of the graph analysis. First, artifact-free epochs are filtered for each frequency band (**A**). Secondly, the functional connectivity matrix based on Phase lag index (PLI) is calculated for each frequency band and epoch (**B**). Finally, Kruskal’s algorithm is applied to obtain the minimum spanning tree (MST) matrix (**C-left**); the resulting loopless graph is displayed on a scalp projection (**C-middle**) and as a tree (**C-right**). The tree view shows the hierarchical structure of the graph starting from an arbitrary root node (in this case, FP1); the color map of the nodes from blue to red represents lower to higher betweenness centrality. For illustrative purposes, this Figure shows the MST obtained from the PLI matrix averaged across epochs and subjects of the control group (N = 15) ([45]).

**Figure 5 behavsci-14-00072-f005:**
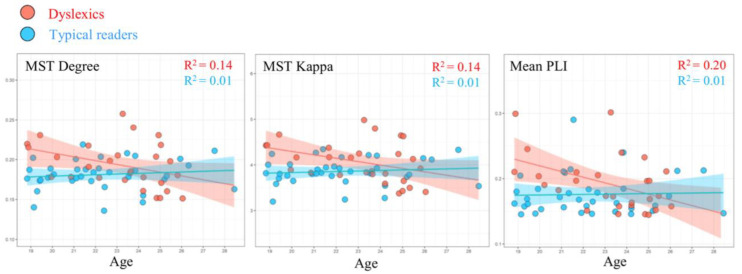
Linear regression showing the relation between age and MST metrics degree and kappa, and mean PLI in the alpha band in typical readers (blue markers) and dyslexics (red markers) [49].

**Figure 6 behavsci-14-00072-f006:**
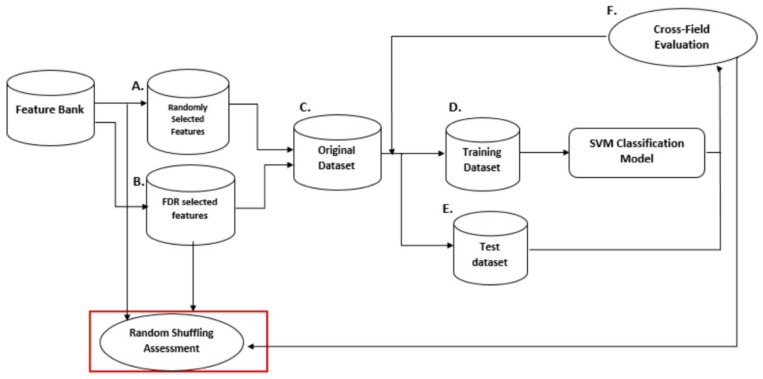
SVM classification and performance assessment by Random Shuffling. We followed two approaches to select features from those available: random selection (**A**) and selection via *t-*tests (**B**) using the original data set (**C**). In both cases, the data set was then divided into a Training set (**D**) and a Test set (**E**) using cross-validation. We assessed each selected feature with the SVM classifier. Finally, a random shuffling cross-fold evaluation (**F**) was performed to ensure the *t-*test-selected features were the most relevant for classification [56].

**Figure 7 behavsci-14-00072-f007:**
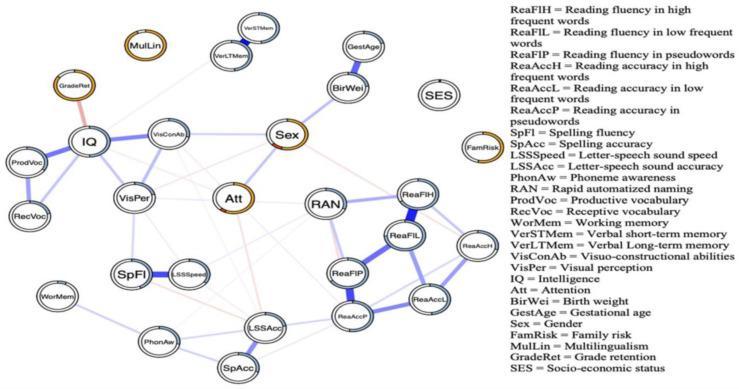
The general network for children with reading difficulties. Positive associations are represented as blue edges, and negative associations are represented as red edges in the network. The width of the edges is proportional to the absolute value of the edge weight. For continuous variables, the blue part of the ring indicates the percentage of explained variance. For binary variables, the orange part of the ring indicates the accuracy of the intercept model, and the red part indicates the additional accuracy achieved by all remaining variables. Hence, the sum of orange and red is the total accuracy of the full model ([57]).

**Figure 8 behavsci-14-00072-f008:**
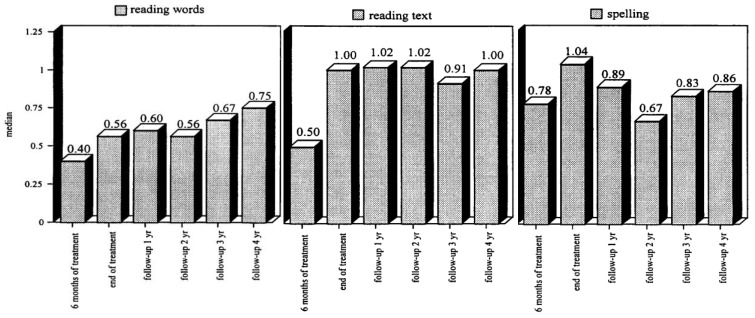
Intervention outcomes during the course of intervention and follow-up. Effect sizes are in terms of a remedial index ranging from zero (nil effect) to one (reading and spelling performance at norm group level) ([92]).

**Figure 9 behavsci-14-00072-f009:**
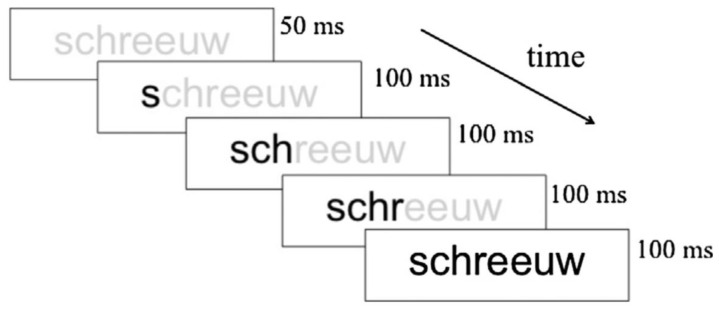
Example of a practice item from the reading training based on letter–speech sound mapping. The presentation of the word schreeuw [sxre!u] (English: shout) under time-demanding conditions. The visual presentation is sound by sound: s[s] _ ch[x] _ r[r] _ eeuw[e!u]. (IPA symbols in brackets) [103].

**Figure 10 behavsci-14-00072-f010:**
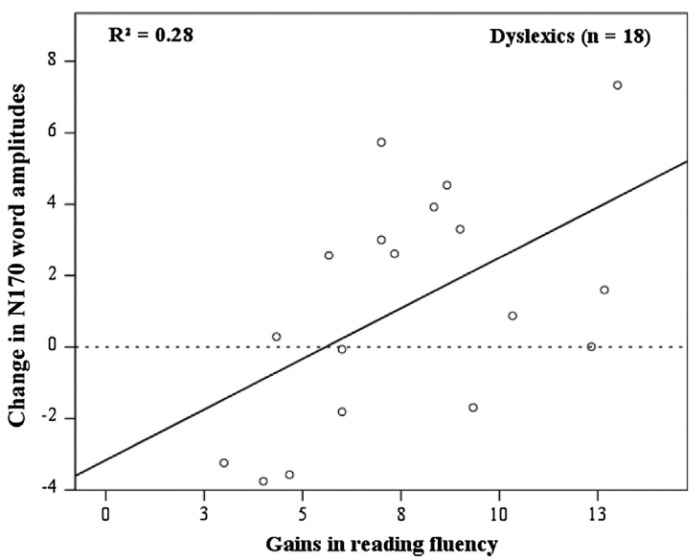
Linear regression between post-pre-test changes in N170 amplitudes to words at the left posterior electrodes (average of P9, P7, PO7, and O1) and gains in reading fluency. Note that a change towards positive values along the *y*-axis refers to a decrease in N170 amplitude [103].

**Figure 11 behavsci-14-00072-f011:**
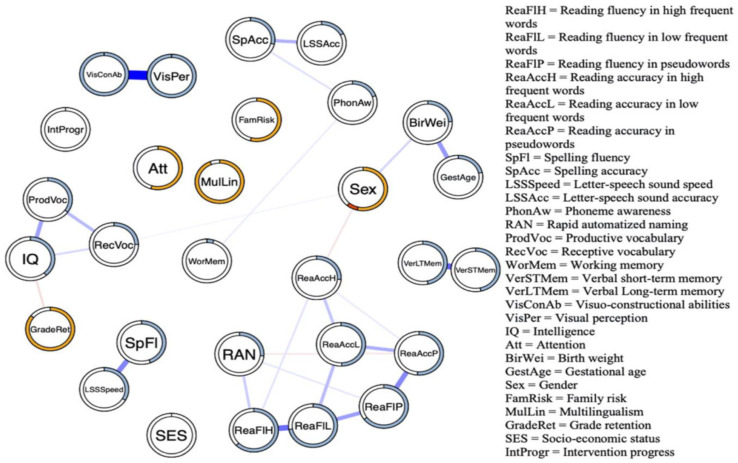
Network displaying the relationships between variables in the framework of reading disabilities. Only children who received reading intervention were included in this sample. Intervention progress was included as a moderator. Positive associations are represented as blue edges, and negative associations are represented as red edges in the network [57].

**Table 1 behavsci-14-00072-t001:** Top-10 of the most productive Dutch authors on Dyslexia. The number of articles (n.a.) and the number of citations (n.c.) are based on a Web of Science search (date: 15 March 2023) with the term ‘dyslexia’ in “All Fields” and last name plus first initial in “Author”.

Author	Affiliation	n.a.	n.c.
Verhoeven, L.	Radboud University	60	669
De Jong, P.F.	University of Amsterdam	51	1088
Van der Leij, A.	University of Amsterdam	46	900
Segers, E.	Radboud University	36	342
Fisher, S.E.	Radboud University	35	911
Tijms, J.	IWAL/University of Amsterdam	31	413
Blomert, L.	University of Maastricht	30	1491
Bonte, M.	University of Maastricht	25	354
Maassen, B.	Radboud University	24	499
Van Bergen, E.	Vrije Universiteit, Amsterdam	21	781

**Table 2 behavsci-14-00072-t002:** Summary of minimum spanning tree (MST) measures using EEG graph analysis [45].

*N*	Nodes	Number of nodes in the MST
*m*	Links	Number of links in the MST
	Degree	Number of neighbors for a given node in the
		MST
*L*	Leaf fraction	Fraction of nodes with degree = 1 (leafs) in the
		MST
*d*	Diameter	Largest distance between any two nodes of the
		tree
	Eccentricity	Longest distance between a reference node
		and any other node
*BC*	Betweenness	Fraction of all shortest paths that pass through
	centrality	a particular node
*k*	Kappa	Measure of the broadness of the degree
		distribution (degree divergence)
*Th*	Tree hierarchy	A hierarchical metric that quantifies the trade-
		off between the large-scale integration in the
		MST and the overload of central nodes
*R*	Degree	Correlation between the degrees of a node and
		the degree of correlation neighboring vertices
		to which it is connected

**Table 3 behavsci-14-00072-t003:** Performance of the classification models. The complexity of the model is reduced from left to right. SVM = Supervised Vector Machine; k-NN = k-Nearest Neighbors [56].

Classifier	SVM Polynomial Kernel	SVM Linear Kernel	k-NN, k = 7	k-NN, k = 3
Accuracy	90.96	95.34	86.04	81.39
Sensitivity	90.00	96.42	86.66	85.71
Specificity	92.30	93.33	84.61	73.33
Precision	96.42	96.42	92.85	85.71

## Data Availability

Issues regarding the data of our studies reviewed should be addressed to the first authors of the pertinent study.

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
