# Peer review of "Dyslexia, the Amsterdam Way"

_behavsci, 2024, doi:10.3390/bs14010072_

Round 1
Reviewer 1 Report
Comments and Suggestions for Authors
You can find my views in the attached file.

Minor editing of English language required
Author Response
Response to Reviewer 1
Clarification
We are grateful for the time and effort of the reviewers invested in commenting on our manuscript. Before responding to the comments, a clarification is in order. Our manuscript is not a research report. It is a review of some of the work we conducted in our dyslexia research. We have been asked to contribute to a special issue devoted to dyslexia research in the International Journal of Environmental Research and Public Health (IJERPH). The issue editor asked explicitly for a contextual review of our research. Accordingly, the current ms has been written in close agreement with the issue editor. Following submission, however, the managing editor informed us that ‘Clarivate has discontinued coverage of IJERPH due to the journal failing the Content Relevance criterion’. An editorial board member evaluated the ms prior to peer review and suggested that our ms would fit the scope of the sister journal Behavioral Sciences. Together with the special issue editor, we swallowed our frustration and agreed to submit to Behavioral Sciences. We hope that this will clarify the review-type format and goal of the current manuscript.
Comment 1: Our ms is not a data report but a review of some of the work we performed on dyslexia (see clarification above). We followed the suggestions by reducing the length of the Introduction, Context, and Understanding sections of the ms.
Comment 2: We followed the suggestion of the reviewer by inserting signposts along the way to facilitate the reading of the longer sections of the ms.
Comment 3: We considered but rejected the suggestion of the reviewer to include a Methodology section. In reviewing our work, we included methodological details of each of the studies reviewed. These details should be sufficient for an independent understanding of the study. Interested readers may then turn to the published studies for the methodological nitty-gritty.
Comment 4: The Introduction includes a clear statement of the goal of the ms. This goal is to review some of our work on dyslexia with a special emphasis on studies aimed at providing a deeper understanding of reading difficulty, investigating tools that are helpful in assessment (dynamic testing, artificial script, biomarkers), and intervention (neural correlates, reading network changes). The specific goal of each study reviewed is clearly stated in the ms.
Comment 5: We believe that the reviewer fails to do justice to the ms in this regard. The ms does not claim to provide an overall view on dyslexia including all potentially relevant facets of reading difficulty. The goal is to review some of our work. The LEXY intervention is applied widely in practical settings in the Netherlands as is the dynamic assessment involving the reading of artificial script. With the advance of improved EEG methodology, assessment and evaluation of intervention benefits electrocortical physiology is on the verge of being applied in practical settings. Fraga Gonzalex et al., (2021) illustrates a step in this direction by combining artificial script learning in a practical setting and EEG activation to artificial script in a laboratory setting.
Comment 6: Here the reviewer addresses a number of issues that are important but far beyond the current ms. The ms does not pretend solving all these issues. With regard to the LEXY program and dynamic assessment using artificial script it can be concluded that these procedures are feasible in practical settings and successfully applied with long term effects. We don’t deny that the practical implications of network analysis but do want to indicate that network analysis has been used successfully in the evaluation of intervention outcomes in clinical studies (e.g., https://doi.org/10.1159/000495045).
Comment 7: Again, the reviewer addresses important issues that cannot be solved in a single ms that, in addition, is targeted to the work performed in a single laboratory/center. We started out by adopting the prevalent notion of dyslexia as a neurobiological disorder. During the course of our work, we are not so sure. This hesitation is highlighted in the final section of the ms pointing to some issues involved in the neurobiological analysis of reading difficulty and the thorny category-dimension issue of the phenomenon. Finally, the ms is written for peers. Educators and clinicians might find the information presented of interest to them but we did not target the ms to an applied audience.
Reviewer 2 Report
Comments and Suggestions for Authors
The manuscript appears to be a compilation of studies on dyslexia conducted at the Department of Psychology at the University of Amsterdam. However, it's unclear what precisely is being reviewed in this paper. Are the authors evaluating the research activities of the Department of Psychology at UvA, or are they summarizing a series of studies on a specific aspect of dyslexia? It's essential for readers to have a clear understanding of the paper's focus and its contribution to the field of dyslexia.
Currently, the objectives, hypotheses, or central claims of this study are not well-defined and may leave readers puzzled about its purpose. To make this paper more relevant to those working in the field of dyslexia, it is crucial to articulate a clear research question or hypothesis and align the subsequent discussion and findings accordingly.
Furthermore, the manuscript gives the impression of being more like a chapter from a book than a standalone review article. It would be beneficial for the authors to explicitly state the type of review being conducted and describe the methodology employed, ensuring that the work's replicability is transparent.
The paper lacks clarity in terms of context, making it challenging for readers to grasp the significance of the information presented, including the tables and figures. It is important to connect these elements cohesively to enhance comprehension.
Additionally, certain details, such as information about lab members and their roles, appear to be irrelevant to the central theme of the review. Consider removing extraneous information to streamline the paper.
Moreover, the current version is excessively lengthy and covers a wide range of topics. A more concise approach is advisable. I suggest that the authors formulate a broader research question or hypothesis related to dyslexia and structure their discussion to present findings from their studies in a more concise and focused manner.
Author Response
Response to Reviewer 2
Clarification
We are grateful for the time and effort of the reviewers invested in commenting on our manuscript. Before responding to the comments, a clarification is in order. Our manuscript is not a research report. It is a review of some of the work we conducted in our dyslexia research. We have been asked to contribute to a special issue devoted to dyslexia research in the International Journal of Environmental Research and Public Health (IJERPH). The issue editor asked explicitly for a contextual review of our research. Accordingly, the current ms has been written in close agreement with the issue editor. Following submission, however, the managing editor informed us that ‘Clarivate has discontinued coverage of IJERPH due to the journal failing the Content Relevance criterion’. An editorial board member evaluated the ms prior to peer review and suggested that our ms would fit the scope of the sister journal Behavioral Sciences. Together with the special issue editor, we swallowed our frustration and agreed to submit to Behavioral Sciences. We hope that this will clarify the review-type format and goal of the current manuscript.
Comment 1. The major goal of the ms is to review some of the work we (Section Developmental Psychology, University of Amsterdam) performed on dyslexia. This goals and the specific aims are now stated clearly in the Introduction (focus on ‘understanding dyslexia’, its “assessment’, and ‘intervention’ strategies).
Comment 2. The presentation of each study includes a clear goal (research question) of the study, necessary details of the methodology employed, the major findings that emerged from the study, the interpretation of the data, and their wider implications. These specifics differ across studies, but the general goals are stated in the Introduction (understanding, assessment, intervention).
Comment 3. The overarching goal is a review of some of our own work on dyslexia. The ms is not intended to provide a review of dyslexia research nor is it a data-oriented report (see clarification above).
Comment 4. We believe that we did our utmost in presenting the material in a concise and clear fashion, including Tables and Figures in order to get the significance of the data across.
Comment 5. We reduced the length of the ms but retained the fundamental structure (understanding, assessment, intervention).
Round 2
Reviewer 2 Report
Comments and Suggestions for Authors
Dear Authors,
Thanks for your responses and the diligent efforts you've put into addressing the concerns I raised. Below, you'll find some minor comments that I believe could be beneficial.
Best,
Minor Suggestions:
1. Line 34: Eliminate the preposition "in"
2. Lines 78-80: Include a reference.
3. Figure 1: Remove the caption labeled "Figure 3."
4. Figures: Enhance resolutions and consider placing captions below the figures for improved readability. Ensure compliance with copyright regulations if using figures from original studies.
5. Table 2 and some others too: Center the table for better alignment.
6. Figure 5, Line 420: Note that the reference numbers exhibit a jump from 46 to 123.
7. Line 530: "two task" > "two tasks."
8. Lines 577-578: Consider omitting the additional definition of dyslexia, as it may be redundant.
9. Lines 1214, 1230, 1251: Remove unnecessary spaces.
Author Response
See file.
